# Analysis on the Choice of Livelihood Strategy for Peasant Households Renting out Farmland: Evidence from Western Poverty-Stricken Areas in China

**Jie Cai** [1] , **Ting Wang** [2] **, Xianli Xia** [3,*] **, Yazhi Chen** [1] **, Hongqiang Lv** [1] **and Ni Li** [1]

1   School of Public Administration and Law, Chang'an University, Xi'an 710064, China;
    wh_caijie@126.com (J.C.); yzchenn@chd.edu.cn (Y.C.); hqlv369@chd.edu.cn (H.L.); lini366@sina.com (N.L.)
2   International Education College, Kedagaoxin University, Xi'an 710109, China; swufewangting@163.com
3   College of Economics and Management, Northwest A & F University, Yangling 712100, China
*   Correspondence: xnxxli@nwsuaf.edu.cn; Tel.: +86-298-708-2420

**Abstract:** Investigating the choice of livelihood strategies has great significance for improving the living standards of peasant households who rent out farmland. This study evaluates the impact of renting-out land on households' livelihood strategies in China's western poverty-stricken areas. Data were obtained from cross sectional survey of 585 field survey data from peasant households who rent out land. The K-means clustering method was used to classify the livelihood strategies of the sample households. In view of sustainable livelihood framework, this paper used combination weighting model based on game theory to calculate the quo of households' livelihood capital. The Multinomial Logistic Regression was used to explore the relationship between livelihood capitals and livelihood strategies. Results show that: livelihood strategy of households who rent out the land can be divided into "agricultural-led" livelihood strategy, "working-oriented" livelihood strategy and "part-time" livelihood strategy. Additionally, the results of Multinomial Logistic Regression show that the households with high human capital and financial capital tend to choose the "working-oriented" livelihood strategy and the households with high natural capital tend to choose the "agricultural-led" livelihood strategy. Therefore, in order to realize the sustainable livelihood of these households, different policy support should be proposed based on the heterogeneity of households in the process of land transfer.

**Keywords:** households who rent out land; livelihood strategy; livelihood capital; western poverty-stricken areas in China

---

## 1. Introduction

Since the reform and opening up, the transfer of land contractual management rights in China have experienced a process from prohibition to acquiescence to encouragement. The land transfer market has shown a booming trend. The land transfer rate increased from 8% in 2008 to 31.4% in 2015 [1]. Under the background of the land resources loss and the growing population, reasonable land transfer has an important impact on promoting the scale operation of agricultural land, improving the land production efficiency and developing modern agriculture [2]. Studies have shown that land resources are still the most important livelihood capital for farmers in western poverty-stricken areas. For farmers, the rented-out land means the risk buffer function of land is lost. If the households who rent out the land cannot find a suitable livelihood strategy, the interests of these farmers will be damaged to some extent. The sustainable livelihood analysis framework is widely used in the study of poverty-related problems because it can explain the complexity of the living conditions of poor farmers

and the strategic nature of their living safety behaviours [3,4]. How to realize the transformation of livelihood strategies of poor farmers in poverty-stricken areas under the new situation is the key to effectively lifting poverty. In fact, the choice of farmers' livelihood strategies is not free and subject to multiple constraints of economy, society and assets [5]. Usually, rational households who rent out the land tend to choose a high-return livelihood strategy. But subject to resource endowments and social systems, some farmers choose a low-return livelihood strategy. While, the existing research rarely has enough evidence to support that the households who rent out the land can choose the livelihood strategy reasonably. In the current risk environment caused by factors such as market, system, policy and nature, the livelihood capital owned by households can affects the farmers' livelihood strategy choices directly [6,7]. Generally speaking, households with more physical capital and natural capital tend to choose agricultural-led livelihood strategies. While households who have more social capital, human capital and financial capital are more likely to choose other non-agricultural-led livelihood strategies [8,9].

The western region is the most important poverty-stricken areas in China. In this region, people face many challenges such as deterioration of ecological environment, shortage of water resources and lag in social and economic development. In the "China Rural Poverty Alleviation and Development Program (2011–2020)," the government has identified 14 concentrated contiguous poverty areas and there are 9 concentrated contiguous poverty areas in the western region. Although the western poverty-stricken areas have also achieved certain development since the reform and opening up, the income gap among eastern, central and western regions has been continuously expanding and the poor population is further concentrated in the western region. In the western poverty-stricken areas of China, the small-scale peasant economy is the main source of income, so farmers in these areas are the groups with the deepest poverty and it is most difficult for these farmers to get rid of poverty [10]. Due to the lack of funds and technical support, agricultural production is not conducive to increasing income. Therefore, famers in western poverty-stricken areas participate in land transfer can obtain rents and be liberated from the land and transfer to non-agricultural economic activities with relatively high labour returns, then the rural distribution structure could be improved [11]. From the observation of the past 10 years, the land transfer in poverty-stricken areas has formed a certain scale but the participation rate is still low which is subject to exogenous factors such as institutions and policies and endogenous factors such as farmer households and resource endowments and characteristics of farmland transfer behaviour. Farmland is the most important livelihood asset for farmers in poverty-stricken areas. However, due to the difference in living patterns caused by human-land relations, different farmers have different behavioural capabilities and choices for land. Therefore, it is of great significance to analyse the impact of the choice of livelihood strategy for households who rent out land in western poverty-stricken areas. On one hand, it can clarify the influence mechanism of households' livelihood capital on the choice of livelihood strategy from the micro perspective of farmers and on the other hand, it can help to formulate effective land transfer incentive measures and supplement the shortage on the choice of livelihood strategy.

## 2. Analysis Framework

In this paper, we assume that the households who rent out the land could adopt a range of livelihood activities to maximize their income under the established livelihood capital conditions. In theory, rational households tend to choose high-return livelihood strategies. But in fact, some farmers choose low-return livelihood strategies subject to resource endowments and social systems. Accurately identifying the selection process and causes of livelihood activities of households who rent out farmland is of great significance to the country and the government to formulate effective policy intervention mechanisms. Assume that *i*th household who rent out the land can make the optimal livelihood strategy choices based on their own livelihood capital. Then these households could have the greatest utility. The theoretical formulas are as follows:

$$\begin{cases} y_i = F(A_i) + \varepsilon_i \\ y_i' = F'(A_i) > 0 \end{cases} \tag{1}$$

where $i$ ($i = 1, 2, \ldots, N$) is the households who rent out the land; $y_i$ is the combination of a range of livelihood activities of the households who rent out the land; $A_i$ is the livelihood capital of the households who rent out the land; $\varepsilon_i$ is the disturbance terms; $y_i$ is an increasing function of the households' livelihood capital. Generally speaking, the allocation of livelihood capital by the family members of the households is the process of selecting the livelihood activities. Different households have different choices in their livelihood activities which will make the households choose the different livelihood strategies. If the benefit from each livelihood activity is $p_i$, the total income of the household is:

$$Y = \sum_{i=1}^{n} p_i y_i \tag{2}$$

In order to maximize the utility of the livelihood capital, the household needs to optimize the allocation of existing resources to obtain the optimal configuration combination. The theoretical formulas are as follows:

$$\begin{cases} y_i = \max_{A_i} U \sum_{i=1}^{n} p_i y_i = \max_{A_i} U(\sum_{i=1}^{n} p_i(F(A_i) + \varepsilon_i)) \\ s.t \sum_{i=1}^{n} A_i \leq A_0 \end{cases} \tag{3}$$

where $A_0$ is the maximum of livelihood capital on the household who rent out the land.

## 3. Materials and Methods

### 3.1. Data Collection

Data were obtained from cross sectional household survey in China's poverty-stricken areas from November 2015 to May 2016. In land transfer system, data were collected from household who participated in land transfer and not participated in land transfer. In order to ensure the representativeness of the survey sample, this survey relied on the criteria of stratified sampling and the first-level sample unit was selected according to the sampling method proportional to the probability and scale. The samples mainly came from Longxi county, Gangu County and Maiji District of Gansu Province, Pengyang County and Yuanzhou District of Ningxia, Yongshou County and Chunhua County of Shaanxi Province. We did face-to-face interviews with the head of household or those in charge. This gave a total number of 590 households who rent out the land and we conducted a logical check and interval check on the data, then excluded the questionnaire samples containing the missing values. Finally, data from 585 households have been used in the analysis. The questionnaire mainly consists of three parts. The first part is whether the households rent out of the land which is used to screen study subjects. The second part is the households' labour force and their income which are used to analyse the choice of households' livelihood strategies. The third part is the households' living capital which is used to analyse the factors affecting the choice of livelihood strategies.

### 3.2. Combination Weighting Model Based on Game Theory

Subjective weighting method and objective weighting method are the two main methods for measuring index weights. Subjective weighting method determines weight based on evaluation of the importance on indicators. Objective weighting method determines the weight according to the size of the original information. Subjective weighting method can reflect the intentions of decision makers better but it is too subjective. Objective weighting method starts from objective data and relies on complete mathematical theory and methods but it ignores the real situation [12]. The combination weighting model based on game theory use game theory to seek consistency and compromise between

the subjective and objective weights, that is, to minimize the deviation between the subjective and objective weights. Then it can maximize the common interests.

(1) Subjective weight based on Delphi method

In order to improve the scientific nature of the empowerment of livelihood capital indicators, this paper selects Delphi method which include expert score and classic literature to calculate subjective weights. Based on the classic literature, we constructed the index system. Through multiple anonymous scoring of senior experts in the field of livelihood capital research, we determined the subjective weights of each indicator in the indicator system. The weight vector obtained by the subjective weighting method based on the Delphi method is denoted as $u_1$.

(2) Objective weight based on entropy method

Due to the difference in the magnitude of each indicator, we should standardize the data firstly. In order to eliminate the influence of negative values on the logarithm latter, we standardize the standardized data again. Then, we get the standardized data $x_{ij}$.

Secondly, we unify the standardized data and calculate the entropy of the corresponding indicators of each dimension, as follows:

$$H_j = -k \sum_{i=1}^{n} f_{ij} \ln f_{ij} \tag{4}$$

where $f_{ij} = x_{ij} / \sum_{i=1}^{m} x_{ij} (i = 1, 2, \ldots, n)$, $k = 1/\ln n$;

Thirdly, we evaluate the entropy weight of each index. Then the weight vector obtained by the entropy method is denoted as $u_2$:

$$w_j = \frac{1 - H_j}{m - \sum_{j=1}^{m} H_j} \tag{5}$$

(3) Combined weight model

We calculated the five dimensions of livelihood capital which include human capital, material capital, natural capital, financial capital and social capital by combination weighting model based on game theory.

The possible weight combination of the subjective weight vector $u_1$ and the objective weight vector $u_2$ can be expressed as:

$$U = \sum_{K=1}^{2} \alpha_k u_k^T \quad (\alpha_k > 0) \tag{6}$$

where $\alpha_k$ is linear combination coefficient ($k$ = 1,2).

Then, we find the best weight between subjective weight and objective weight u drawing on the idea of game theory and make sure that the deviation among u, $u_1$ and $u_2$ is minimized. The optimal objective function is denoted as follows:

$$\min \| \sum_{k=1}^{2} \alpha_k \times u_k^T - u_i^T \|_2 \quad (i = 1, 2) \tag{7}$$

According to the differential properties of the matrix, we can see that the optimal condition of the first derivative of Equation (7) is denoted as follows:

$$\sum_{k=1}^{2} \alpha_k \times u_i \times u_k^T = u_i \times u_i^T \quad (i = 1, 2) \tag{8}$$

The corresponding equations of Equation (8) are:

$$\begin{bmatrix} u_1 \cdot u_1^T & u_1 \cdot u_2^T \\ u_2 \cdot u_1^T & u_2 \cdot u_2^T \end{bmatrix} \begin{bmatrix} \alpha_1 \\ \alpha_2 \end{bmatrix} = \begin{bmatrix} u_1 \cdot u_2^T \\ u_2 \cdot u_2^T \end{bmatrix} \quad (i = 1, 2) \tag{9}$$

According to Equation (9), we can calculate the value of $(\alpha_1, \alpha_2)$. We use $\alpha_k^* = \alpha_k / \sum_{k=1}^{2} \alpha_k$ to normalize the value of $(\alpha_1, \alpha_2)$. The final combined weight of each indicator can be expressed as:

$$u^* = \sum_{K=1}^{2} \alpha_k^* u_k^T \tag{10}$$

### 3.3. K-Means Clustering Analysis

The essence of cluster analysis is to divide the data into several categories according to the distance. In this method, we should narrow the difference of data within the category and to expand the difference of data between the categories. In this paper, the K-means clustering analysis method is used to classify the households who rent out the land. According to the households' characteristics, the sample data were divided into K categories as the initial cluster centre. Based on this, we calculated the distance of each sample to the cluster centre separately and classified the samples into the categories according to the nearest distance principle. In order to get a new cluster centre we measured the average of each newly formed cluster data according to the newly formed central location. If the cluster centres of two adjacent cities are the same, we could insist that the sample reaches a certain convergence criterion. Otherwise we must continue to adjust the sample data until the sample data are correctly classified.

### 3.4. Multinomial Logistic Regression

Since the livelihood strategy of households who rent out the land is a discrete variable, multinomial logistic regressions are used to estimate the influencing factors of these households' livelihood strategy choice. We suppose that the *i*th household who rent out the land can choose from J kinds of mutually exclusive livelihood strategies. The random utility function brought by *i*th household who rent out the land chooses *j*th livelihood strategy can be expressed as:

$$U_{ij} = x_i \beta_j + \varepsilon_{ij} \tag{11}$$

where $x_i$ include the variables of livelihood capital and other characteristics on the *i*th household who rent out the land, $\varepsilon_{ij}$ is the random disturbance. Usually rational households who rent out the land will choose the most effective solution from the j kinds of livelihood strategies.

The probability that the ith household who rent out the land chooses the *j*th livelihood strategy is denoted as follows:

$$P_j = P(U_{ij} \geq U_{ik}, \forall k \neq j) = \frac{\exp(x_i \beta_j)}{\sum_{k=1}^{4} \exp(x_i \beta_k)} \tag{12}$$

## 4. Results

### 4.1. The Livelihood Capital of Households Who Rent Out the Land

On the basis of ideas on sustainable livelihoods, Chambers et al. [13] defined livelihoods as a means of earning a living based on capabilities, assets which include reserves, resources, claims and enjoyment and activities [14]. With the in-depth study of rural poverty, the British International Development Agency proposed the Sustainable Livelihood Analysis Frame work in 2000. The DFID model makes human capital, natural capital, physical capital, financial capital and society as the core of the study [15,16]. With reference to the DFID model, combined with the social and economic development of the western poverty-stricken areas and the households' livelihood characteristics, this

paper divides the livelihood capital of households who rent out the land into five categories: human capital, natural capital, physical capital, financial capital and social capital. On this basis, we designed the index system for measuring the livelihood capital of households who rent out the land in the study area (Table 1). From the Table 1, we can see:

(1) Human capital indicators and calculations.

Human capital is the main reason for promoting the growth of the national economy today. Schultz [17] believed that population quality and knowledge investment determined the future prospects of mankind largely. For farmers, human capital mainly includes individual knowledge, skills, abilities and health status [8]. The quantity and quality of human capital determine whether farmers can use other capital [18]. Refers to the research of Sharp [19], Brown et al. [20], Wu et al. [21] and Zhang et al. [22], we select the four variables including the number of labour, the proportion of labour, the years of education for adults and the health of family members to measure the human capital of households who rent out the land. The specific meanings of the variables are shown in Table 1. In the process of measurement, we give the weights of 0.3724: 0.2167: 0.2100: 0.2009 for the four indicators.

(2) Natural capital indicators and calculations

For farmers, land resources are the most important natural assets of their families. Households usually dispose of their land resources reasonably according to their family living conditions. Refers to the research of Sharp [19], Li et al [18], Yang and Zhao [23], Su and Shang [8], we select the two variables including the actually allocated land area and the operating land area to measure the natural capital of households who rent out the land. The specific meanings of the variables are shown in Table 1. In the process of measurement, we give the weights of 0.5749: 0.4251 for the two indicators.

(3) Physical capital indicators and calculations

For farmers, physical capital mainly includes production tools and infrastructure for daily production and life [24]. Different households face basically the same infrastructure but there is a big difference between productive and consumer infrastructure. Refers to the research of Nelson et al. [25], Cai [26], Duan et al. [27], we select the two variables including house value and agricultural machinery value to measure the physical capital of households who rent out the land. The specific meanings of the variables are shown in Table 1. In the process of measurement, we give the weights of 0.5796: 0.4204 for the two indicators.

(4) Financial capital indicators and calculations

Financial capital mainly refers to the cash that farmers can control and raise. Usually, the financial capital includes the households' income, raised funds from formal or informal channels and free aid fund. Refers to the research of Li et al. [18], Zhang et al. [28], Zhu et al. [29], we select the four variables including per capital net income, the availability to get free aid, loan obtained from the formal channel and loan obtained from the informal channel to measure the financial capital of households who rent out the land. The specific meanings of the variables are shown in Table 1. In the process of measurement, we give the weights of 0.4771: 0.1193: 0.1105: 0.2931 for the four indicators.

(5) Social capital indicators and calculations

Social capital mainly refers to the social relations and social networks that households who rent out the land use to maintain sustainable livelihoods. Refers to the research of Zhao and Xue [30], Ding et al. [31], Ning [32], we select the four variables including participation in public affairs, the size of the New Year's network, political relations between relatives and friends, annual transportation and communication fees to measure the social capital of households who rent out the land. The specific meanings of the variables are shown in Table 1. In the process of measurement, we give the weights of 0.2382: 0.0811: 0.3381: 0.3426 for the four indicators.

**Table 1.** Livelihood capital indicator assignment and weight.

| Index | Variable | Definition | Delphi Method | Entropy Weight Method | Combination Weighting Model |
|---|---|---|---|---|---|
| Human capital | the number of labour | The number of labour owned by the households (infants, children in elementary school equal 0, disabled and unemployed elderly = 0; children in junior high school = 0.3; children in high school= 0.6; elderly who can only work part-time = 0.5; adult labour = 1 | 0.4000 | 0.2245 | 0.3724 |
| | the proportion of labour | The proportion of labour force to the total population of the household | 0.2000 | 0.2672 | 0.2167 |
| | the years of education for adults | Average years of schooling completed by household members older than 18 years | 0.2000 | 0.2319 | 0.2100 |
| | the health of family members | The average health status of family members (the individual health status is: cannot take care of themselves=1; have disease but can take care of themselves = 2; general = 3; healthy = 4; more healthy = 5 | 0.2000 | 0.2764 | 0.2009 |
| Natural capital | the actually allocated land area | The actual value of the household's land area (1/15 hectare) | 0.6000 | 0.4728 | 0.5749 |
| | the operating land area | Actual value of cultivated farmland area (1/15 hectare) | 0.4000 | 0.5272 | 0.4251 |
| Physical capital | house value | The house's discounted value combined with the structure, structure, cost of construction, age of use and so forth, converted into the present value of RMB (yuan, logarithm) | 0.6000 | 0.4801 | 0.5796 |
| | agricultural machinery value | The agricultural machinery's discounted value which includes tractor (tractor, tricycle), agricultural machinery (pump, diesel, rice transplanter, harvester, seeder, thresher, tiller, rotary tiller, et al.) (yuan, logarithm) | 0.4000 | 0.5199 | 0.4204 |
| Financial capital | per capital net income | Per capital net income of the household(yuan, logarithm) | 0.4000 | 0.5141 | 0.4771 |
| | the availability to get free aid | Weather can get free assistance from the government, relatives or friends? (yes = 1; no = 0) | 0.1000 | 0.1267 | 0.1193 |
| | loan obtained from the formal channel | The amount of loans from formal institutions such as banks and credit unions (yuan, logarithm) | 0.1000 | 0.1162 | 0.1105 |
| | loan obtained from the informal channel | Amount of loans from informal institutions such as relatives, friends, neighbours or usury (yuan, logarithm) | 0.4000 | 0.243 | 0.2931 |
| Social capital | participation in public affairs | participation in village's public affairs(Never = 1; occasionally = 2; general = 3; more = 4; often = 5) | 0.2000 | 0.2404 | 0.2382 |
| | the size of the New Year's network | The size of relatives and friends need to visit in the Chinese New York. | 0.3000 | 0.0667 | 0.0811 |
| | political relations between relatives and friends | Are there any civil servants or village officials among relatives and friends? (yes = 1; no = 0) | 0.3000 | 0.3401 | 0.3381 |
| | annual transportation and communication fees | Transportation and communication expenses for actual household expenses (yuan, logarithm) | 0.2000 | 0.3528 | 0.3426 |

### 4.2. The Classification of Livelihood Strategy on Households Who Rent Out the Land

The livelihood strategy is a livelihood activity based on the choice of livelihood capital elements. It refers to the combination of activities and choices made by the family to achieve sustainable livelihoods. Various livelihood activities are combined and promoted each other under different living capital conditions and finally formed a livelihood strategy. The choice of livelihood strategy reflects the extent of households' utilization of livelihood capital and the livelihood activities. Usually, a reasonable and appropriate choice of livelihood strategies can achieve the goal of sustainable livelihood. Considering that different scholars have different research objectives and research background, they divide the livelihood strategies in different ways. Scoones [33] divides livelihood strategies into two types: single livelihood strategies that rely on agricultural production and diversity livelihood strategy. Li et al. [34] divides households' livelihood strategies into four categories: agricultural and forestry-based livelihood strategies, livestock farming livelihood strategies, non-agricultural management livelihood strategy and migrant work-based livelihood strategy. Li [35] classifies households' livelihood strategies into four categories: traditional livelihood dependence, non-agricultural specialization, subsidy-dependent and diversified livelihood according to a classification method that whether a certain income accounts for more than 50% of total household income; Zhu et al. [29] uses cluster analysis to classify households' livelihood strategies into non-agricultural income-oriented, job-oriented and part-time.

Considering different scholars have different research objectives, they classify the households' livelihood strategies in different ways. Based on the existing research and the availability of data, we select the seven variables which include the proportion of households' member specializing in agriculture to the total labour, the proportion of part-time workers to the total labour, the proportion of specialized workers to the total labour, agricultural income as a share of total household income, income from work and non-agricultural operations as a percentage of total household income, land transfer rental income as a percentage of total household income, other income such as government subsidies as a percentage of total household income as input indicators for cluster analysis. Then we use SPSS to do K-means clustering analysis. The K-means clustering analysis showed that the livelihood strategies of households who rent out the land can be divided into three categories with obvious characteristics. The livelihood characteristics of various types of households are shown in Table 2. We can see that:

**Table 2.** Livelihood strategy classification of households who rent out the land.

| Variable | "Agricultural-Led" Livelihood Strategy | "Working-Oriented" Livelihood Strategy | "Part-Time" Livelihood Strategy |
|---|---|---|---|
| The proportion of households' member specializing in agriculture to the total labour (%) | 78.10 | 24.96 | 26.40 |
| The proportion of part-time workers to the total labour (%) | 4.18 | 3.34 | 60.27 |
| The proportion of specialized workers to the total labour (%) | 9.05 | 71.70 | 8.82 |
| Agricultural income as a share of total household income (%) | 31.80 | 6.55 | 15.33 |
| Income from work and non-agricultural operations as a percentage of total household income (%) | 27.44 | 81.37 | 72.24 |
| Land transfer rental income as a percentage of total household income (%) | 15.26 | 6.00 | 6.58 |
| Other income such as government subsidies as a percentage of total household income (%) | 25.50 | 6.07 | 5.85 |
| Number | 196 | 234 | 155 |
| Ratio (%) | 33.50 | 40.00 | 26.50 |

Livelihood strategy 1: There are 196 households who rent out the land belonging to this type, accounting for 33.50% of the sample farmers. In terms of employment, the proportion of households' member specializing in agriculture accounts for the largest proportion of household labour, with an average of 78.10%. In terms of income, agricultural income accounts for the largest proportion of total household income, with an average of 31.80%. Other income such as government subsidies accounts for 25.50% of total household income. Then this paper defines it as "agricultural-led" livelihood strategy.

Livelihood strategy 2: There are 234 households who rent out the land belonging to this type, accounting for 40.00% of the sample farmers. In terms of employment, the proportion of specialized workers accounts for the largest proportion of household labour, with an average of 71.70%. In terms of income, income from work and non-agricultural accounts for the largest proportion of total household income, with an average of 81.37%. Then this paper defines it as "working-oriented" livelihood strategy.

Livelihood strategy 3: There are 155 households who rent out the land belonging to this type, accounting for 26.50% of the sample farmers. In terms of employment, the proportion of part-time workers accounts for the largest proportion of household labour, with an average of 60.27%. In terms of income, income from work and non-agricultural accounts for the larger proportion of total household income, with an average of 72.24%. Then this paper defines it as "part-time" livelihood strategy.

### 4.3. Analysis of the Influence of Livelihood Capital on the Choice of Livelihood Strategy

We use Multinomial Logistic Regression to analyse the influencing factors of livelihood strategy selection on households' who rent out the land in Stata14.0, the results are shown in Table 3. From the regression results, we can see that the model has a good fitting effect, the likelihood ratio test is 123.41 and $p < 0.01$. This result indicates that at least some of the explanatory variables in the model have a statistically significant effect and the reliability of the model estimation results is high.

**Table 3.** Influencing factors of livelihood strategy selection on households' who rent out the land.

| Variable | $\ln(P_2/P_1)$ | | $\ln(P_3/P_1)$ | | $\ln(P_2/P_3)$ | |
|---|---|---|---|---|---|---|
| | Coefficient | RRR | Coefficient | RRR | Coefficient | RRR |
| Human capital | 6.5734 *** | 715.7768 | 3.6001 *** | 36.6161 | 2.9729 *** | 19.5482 |
| Natural capital | −2.7576*** | 0.0634 | −1.3377 * | 0.1944 | −1.1199 | 0.3263 |
| Physical capital | 0.3798 | 1.4621 | 0.6487 | 1.9131 | −0.3689 | 0.7642 |
| Financial capital | 5.6810 *** | 293.2502 | 0.9773 | 2.6574 | 4.7037 *** | 110.3536 |
| Social capital | 0.3613 | 1.4352 | 0.1759 | 0.8388 | 0.5371 | 1.7110 |
| Control variable | | | Controlled | | | |
| Constant term | 6.0167 * | 410.2149 | 3.7425 | 42.2039 | 2.2742 | 9.7198 |
| LR chi2 | | | 123.41 | | | |
| Prob > chi2 | | | 0.0000 | | | |
| Pseudo R$^2$ | | | 0.0972 | | | |

Notes: *,**,*** indicates level of significance at the 10%, 5% and 1% level, respectively. The RRR represents the contribution of independent variables to the probability ratio of dependent variables. $\ln(P_2/P_1)$ and $\ln(P_3/P_1)$ are both based on the "agricultural-led" livelihood strategy. $\ln(P_2/P_3)$ is based on the "part-time" livelihood strategy.

The households who rent out the land with high human capital tend to choose the "working-oriented" livelihood strategy. As can be seen from the regression results in Table 3: compared with the "agricultural-led" livelihood strategy, the estimated coefficient of human capital on the choice of "working-oriented" livelihood strategy for households who rent out the land is 6.5734 and passed the 1% significance level test; compared with the "agricultural-led" livelihood strategy, the estimated coefficient of human capital on the choice of "part-time" livelihood strategy for households who rent out the land is 3.6001 and passed the 1% significance level test. That is to say, the households who rent out the land with high human capital tend to choose the "working-oriented" livelihood strategy and the "part-time" livelihood strategy. Compared with the "part-time" livelihood strategy, the estimated coefficient of human capital on the choice of "working-oriented" livelihood strategy for households who rent out the land is 2.9729 and passed the 1% significance level test, that means the households who

rent out the land with high human capital tend to choose the "working-oriented" livelihood strategy. It can be seen that human capital has a significant positive impact on the choice of "working-oriented" livelihood strategies for households who rent out the land. This indicates households who rent out the land with high human capital tend to choose "working-oriented" livelihood strategies. Through in-depth analysis of its influencing factors, we find that with the impact of the sharp decline in the profit margin of agricultural production, labour with higher capacity faces more employment opportunities. Then they are more willing to participate in non-agricultural employment. The households who rent out the land with higher education level can better grasp the economic development situation and employment trend and it is easier for them to grasp the opportunity and maximize the economic value of itself in the process of non-agricultural employment which will help to maximize the economic value for themselves.

The households who rent out the land with high natural capital tend to choose the "agricultural-led" livelihood strategy. As can be seen from the regression results in Table 3: compared with the "agricultural-led" livelihood strategy, the estimated coefficient of natural capital on the choice of "working-oriented" livelihood strategy for households who rent out the land is −2.7576 and passed the 1% significance level test; compared with the "agricultural-led" livelihood strategy, the estimated coefficient of natural capital on the choice of "part-time" livelihood strategy for households who rent out the land is −1.3377 and passed the 10% significance level test. That is to say, the households who rent out the land with high natural capital tend to choose the "agricultural-led" livelihood strategy. Compared with the "part-time" livelihood strategy, the estimated coefficient of natural capital on the choice of "working-oriented" livelihood strategy for households who rent out the land is −1.1199 but it did not pass the significance level test. This indicates households who rent out the land with higher natural capital tend to choose "part-time" livelihood strategies but this impact is not significant. The possible reason is that natural capital is the basis for farmers to carry out agricultural production and households who rent out the land with more agricultural land area and actual cultivated area have higher enthusiasm for participating in agricultural production. These farmers are more willing to participate in agricultural employment. Therefore, natural capital has a significant impact on the choice of "agricultural-led" livelihood strategies for households who rent out the land.

Physical capital has no significant impact on the livelihood strategies choice of households who rent out the land. This may be due to the fact that there is no difference in the fixed assets of the family houses in the three types of livelihood strategies. Due to the poor natural resource endowment in the western poverty-stricken areas, agricultural production has been limited. There is no significant difference among the farm machinery owned by farmers including water pumps, diesel engines, rice transplanters, harvesters, seeders, threshers, micro tillage machines and rotary tillers. Therefore, physical capital has a small impact on the choice of livelihood strategies for households who rent out the land and does not have statistical significance.

The households who rent out the land with high financial capital tend to choose the "working-oriented" livelihood strategy. As can be seen from the regression results in Table 3: compared with the "agricultural-led" livelihood strategy, the estimated coefficient of financial capital on the choice of "working-oriented" livelihood strategy for households who rent out the land is 5.6810 and passed the 1% significance level test; compared with the "agricultural-led" livelihood strategy, the estimated coefficient of financial capital on the choice of "part-time" livelihood strategy for households who rent out the land is 0.9773 but it did not pass the significance level test. That is to say, the households who rent out the land with high financial capital tend to choose the "working-oriented" livelihood strategy and the "part-time" livelihood strategy but the effect of high financial capital on the "part-time" livelihood strategy is not significant. Compared with the "part-time" livelihood strategy, the estimated coefficient of financial capital on the choice of "working-oriented" livelihood strategy for households who rent out the land is 4.7037 and passed the 1% significance level test, that means the households who rent out the land with high financial capital tend to choose the "working-oriented" livelihood strategy. Under the attraction of non-agricultural employment, households with higher per capital annual

income and easier access to free assistance are more willing to choose a "work-oriented" livelihood strategy. Households who can raise more funds including formal channel loans and informal channel loans to obtain credit are more willing to choose "work-oriented" livelihood strategy.

Social capital has no significant impact on the livelihood strategies choice of households who rent out the land. This may be due to the fact that most of the households who rent out the land participate in the land transfer for not a long time and the original kinship and geographical relationship have not changed significantly. That is to say, the social capital of households who rent out the land has not changed much. Therefore, social capital has a small impact on the choice of livelihood strategies for households who rent out the land and does not have statistical significance.

## 5. Discussion and Policy Implication

There is a great debate about whether the households who rent out the land can choose the suitable livelihood strategy in western poverty-stricken areas of China. In this study, we conducted a household investigation of a sample of 585 rural households across seven western poverty-stricken areas in China. We performed descriptive statistical analysis and constructed empirical econometric models. This research generated a rich set of interesting results and thus contributed to a better understanding of the choice of livelihood strategy for households who rent out land livelihoods.

First, K-means clustering analysis results show that the livelihood strategy of the surveyed households who rent out the land can be divided into three categories with obvious characteristics. The three kinds of livelihood strategy are "agricultural-led" livelihood strategy, "working-oriented" livelihood strategy and "part-time" livelihood strategy.

Second, when we used the Multinomial Logistic Regression to analyse the impact of livelihood capital on the choice of livelihood strategies, we found that, human capital, natural capital and financial capital had a significant impact on the choice of livelihood strategies for households who rent out the land. The households with high human capital and financial capital tend to choose the "working-oriented" livelihood strategy. And the households with high natural capital tend to choose the "agricultural-led" livelihood strategy.

The livelihood strategies of households who rent out the land are diverse. Therefore, in order to realize the sustainable livelihood of these households, different policy support should be proposed based on the heterogeneity of households in the process of land transfer. Firstly, the government should provide non-agricultural vocational training for the households who rent out the land. This will help household to improve the competitiveness and job adaptability. Secondly, the Chinese government should better encourage the households to rent out the land to new agricultural business entities and dispose of the livelihood assets of households who rent out the land rationally. Also, the government should provide policy support for households to solve the realization of agricultural production assets and ensure that households who rent out the land can not only obtain stable property income but also obtain non-agricultural income. Thirdly, the credit department should promote diversified services for rural credit supplies, standardize the development of informal microfinance institutions and private finance. These measures will make up for the shortage of funds for households who rent out the land. Finally, it is necessary for agricultural management section to provide better socialized services for households with "agricultural-led" livelihood strategies and encourage them to make the production of high-value crops.

**Author Contributions:** J.C. and X.X. designed the research and wrote the paper. J.C. and W.T. analyzed the data. Y.C., H.L. and N.L. help to modify the paper.

**Funding:** This research was funded by the National Social Science Fund of China grant NO. 17BJY137, Chang'an University Central University Basic Research Business Expenses Special Fund Project grant No. 300102119623.

**Acknowledgments:** The authors appreciate the valuable comments from all anonymous referees as well as the editor of this journal.

**Conflicts of Interest:** The authors declare no conflict of interest.

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
