# Peer review of "Analysis on the Choice of Livelihood Strategy for Peasant Households Renting out Farmland: Evidence from Western Poverty-Stricken Areas in China"

_sustainability, doi:10.3390/su11051424_

Round 1

Reviewer 1 Report

Abstract

Authors need to re-write this section to make the text easily readable and understandable. I suggest the authors give the manuscript to a native English speaker to proofread it. The third line should read data were obtained….

The main results of the logit regression model should be presented. The implications of the findings of the study should also be reported in this section.

Logit regression is used ONLY when the dependent variable is dichotomous (yes/no), (1/0). It is not clear why the authors used this technique given that they didn’t indicate what the specific dependent variable was.

Introduction

This section is generally well-written although authors ought to check grammatical errors. The statement of the problem is not clear. What gaps exist in our current understanding of the behavioral capabilities and choices for land? This then should be the basis of what the authors set out to achieve.

Materials and Methods

Section 3.4 is conceptually flawed. The fact that livelihood strategy is discrete does NOT mean logit regression is appropriate to use. Major assumptions of the Logit model:

·         The dependent variable should be dichotomous in nature (e.g., presence vs. absent; yes or no).

·         There should be no outliers in the data, which can be assessed by converting the continuous predictors to standardized scores, and removing values below -3.29 or greater than 3.29.

·         There should be no high correlations (multicollinearity) among the predictors.  This can be assessed by a correlation matrix among the predictors. Tabachnick and Fidell (2013) suggest that as long correlation coefficients among independent variables are less than 0.90 the assumption is met.

If the dependent variable is “has livelihood strategy or has no livelihood strategy” then the logit model is appropriate. In this instance, it appears there are several categories of livelihood strategy without any hierarchical or numerical order (not ordinal). This indicates that the most appropriate model the authors should use is multinomial regression. Authors should address this major flaw in the work.

An important point is whether the three livelihood strategies are mutually-exclusive or not. For instance, can one respondent be simultaneously involved in Agricultural-led” livelihood strategy, “Working-oriented” livelihood strategy, and "Part-time" livelihood strategy at the same time?

Based on the percentages in Table 2 (do NOT sum up to 100% in rows or columns), it seems to me that respondents answered either yes or no to each of the following: involved in Agricultural-led” livelihood strategy, “Working-oriented” livelihood strategy, and "Part-time" livelihood strategy at the same time? If this suspicion is correct, then the authors must model each livelihood strategy separately using the same predictors.  

Results

This section will change once the authors make the changes I suggested previously

Discussion and Conclusion

This section will change once the authors make the changes I suggested previously

Author Response

Dear Prof. ***

Thank you very much for your letter and comments from referees about our paper submitted to Sustainability (NO.426773).

We have checked the manuscript and revised it according to the comments we submit here the revised manuscript as well as a list of changes.

If you have any question about this paper, please don’t hesitate to let me know.

Reviewer 2 Report

On page five, line 194, I believe if you included a brief discussion onTheodore Schultz's article "Investment in Human Capital" I believe this would be very important to your discussion section in your article.  

Also you said that you used surveys and conducted interviews.  I suggest you might want to tell the reader some of your questions.  Which question/s were designed to elicit a response that answered your research question.  Or which one gave you a surprise response and why?

In your last section, what was the original policy in Western Region of China.  To me that was not clear

Author Response

(The authors gave the same response as above.)

Round 2

Reviewer 1 Report

It appears the authors did not understand my original comments. One of the main issues which they refused to address is the correct name of the statistical model they used to model the relationship between the dependent variable () and the set of predictors. They indicated they used multiple logit regression (which means typically mean many logit models). The dependent variable was coded 1,2,3 etc. with each subgroup as mutually exclusive and without hierarchical order. This implies that the code is nominal. For these reasons, I pointed out that the more appropriate nomenclature is multinomial logistic regression (please see https://newonlinecourses.science.psu.edu/stat504/node/172/).  They refused to correct this error and the explanation they gave is unsatisfactory.

Author Response

Dear Prof. ***

Thank you very much for your letter and comments from referees about our paper submitted to Sustainability (NO.426773).

We have checked the manuscript and revised it according to the comments we submit here the revised manuscript as well as a list of changes.

I apologize for the personal misunderstanding of the original comments and the troubles that caused you some trouble. Taking into account the recommendations of the reviewers and consulting a large number of references, the more appropriate nomenclature is Multinomial Logistic Regression. Then I change the Multiple Logit Regression into Multinomial Logistic Regression.

If you have any question about this paper, please don’t hesitate to let me know.

Round 3

Reviewer 1 Report

All my comments have been addressed